# A Review of Circularly Polarized Dielectric Resonator Antennas: Recent Developments and Applications

**DOI:** 10.3390/mi13122178

**Published:** 2022-12-08

**Authors:** Nur Akmal Abd Rahman, Mohd Najib Mohd Yasin, Imran Mohd Ibrahim, Muzammil Jusoh, Shehab Khan Noor, Mervin Retnadhas Ekscalin Emalda Mary, Norshuhani Zamin, Nurhayati Nurhayati

**Affiliations:** 1Advanced Communication Engineering (ACE) Centre of Excellence, Faculty Electronic Engineering Technology, Universiti Malaysia Perlis (UniMAP), Kangar 01000, Perlis, Malaysia; 2Centre for Telecommunication Research and Innovation (CeTRI), Faculty of Electronic and Computer Engineering, Universiti Teknikal Malaysia Melaka (UTeM), Durian Tunggal 76100, Melaka, Malaysia; 3Department of Information Technology, College of Computing and Informatics, Saudi Electronic University, Riyadh 13323, Saudi Arabia; 4Department of Electrical Engineering, Universitas Negeri Surabaya, Surabaya 60231, Indonesia

**Keywords:** dielectric resonator antenna, feeding technique, circular polarization, unmanned aerial vehicle, millimeter-wave

## Abstract

A comprehensive review on recent developments and applications of circularly polarized (CP) dielectric resonator antennas (DRAs) is proposed in this paper. DRAs have received more considerations in various applications due to their advantages such as wide bandwidth, high gain, high efficiency, low losses, and low profile. A broad justification for circular polarization and DRAs is stated at the beginning of the review. Various techniques such as single feed, dual, or multiple feeds used by different researchers for generating circular polarization in DRAs are briefly studied in this paper. Multiple-input-multiple-output (MIMO) CP DRAs, which can increase channel capacity, link reliability, and data rate, have also been analyzed. Additionally, innovative design solutions for broadening the circular polarization bandwidth and reducing mutual coupling are studied. Several applications of DRA are also discussed comprehensively. This paper finishes with concluding remarks.

## 1. Introduction

The past decade has seen a huge growth in research efforts in communication systems. In a communication system, an antenna plays a crucial role in transmitting and receiving information. Microstrip patch antennas (MPAs) are widely applied due to their light weight, small dimensions, low cost, and easy fabrication, as well as planar geometry [1]. However, at higher frequencies, the MPA suffers from low radiation efficiency due to the inherent metallic losses [2]. Moreover, most designs suffer from a narrow bandwidth, low gain, low axial ratio (AR), and low efficiency [3]. These difficulties have motivated researchers to construct antennas with additional design enhancement in order to increase the bandwidth and performance, as well as reducing their physical size. Dielectric resonator antennas (DRAs) have recently been used to replace MPAs due to their interesting characteristics such as versatility in shape, compact size, zero metallic losses, ease of excitation, and relatively wide impedance bandwidth [4,5,6,7]. Moreover, DRAs exhibit higher radiation efficiency even at higher frequencies due to the absence of intrinsic conductor loss and surface wave loss [8]. DRAs are widely used for high-frequency applications such as radio frequency (RF) energy harvesting [9,10], mm-wave antennas [11,12], global navigation satellite systems (GNSS) [13,14], radar, and biomedical applications.

Different shapes of DRAs have been established since 1980 with the constant work of S. Long [15]. The analysis was performed on a cylindrical DRA. The antenna is fed by a feed probe, as shown in Figure 1. This antenna managed to provide efficient radiation in the direction normal to its ground plane and suitable for millimeter-wave (mm-wave) applications.

The most crucial parameters in antenna are radiation pattern, beam efficiency, directivity, gain, bandwidth, and input impedance. Polarization is also one of the important parameters. Polarization in a given direction is the polarization of an electromagnetic wave transmitted by the antenna. The polarization is stated as the polarization in the direction of maximum gain if the direction is unfixed. According to [16], different polarizations result from different parts of the pattern because the polarization of the emitted energy differs with the radiation from the center of the antenna. Polarization is classified as linear, circular, and elliptical, as shown in Figure 2. These polarizations, depending on the amplitude and phase, shift between two waves. In general, polarization is described by an ellipse. Linear polarization is obtained when the ellipse becomes a straight line; whereas, circular polarization is obtained when the ellipse becomes a circle.

Linear polarization (*T*→∞ or *T* = 0) is realized when the field is corresponded to a direction in time. Linear polarization is obtained if the field vector has one component or two orthogonal linear components with a 180° phase shift [16]. Linear polarization is also associated with an antenna system that is working with vertical and horizontal polarizations. The radiation pattern of an antenna is referred to as vertically polarized if the electric field vector is oscillating in the vertical direction. Conversely, the electric field will oscillate in the horizontal direction for horizontal polarization. For a linearly polarized (LP) wave, *Ex* and *Ey* are in phase with each other; whereas, circular polarization (*T* = 1) occurs when the two orthogonal polarized components (*x*-direction and *y*-direction) have the same magnitude (*Ex* = *Ey*) and a 90° time-phase difference [18].

LP DRAs have been broadly explored and are still receiving considerable interest at present. However, the application of the LP antenna is only capable to detect signals in one direction and will face a polarization mismatch loss when the signal is curving from different angles [19]. Therefore, a circularly polarized (CP) antenna is preferable for various applications because it can receive a component of the signal persistently due to the angular variation, regardless of receiver orientation, thus making the CP antenna capable to transmit and receive signals in all planes so that the strength of the signals is not lost and transferred to another plane and is still used [20]. As a result, the gain and performances of the system can increase and the multipath effect can improve [21,22]. Moreover, CP antennas have received considerable attention due to their advantages such as immunity to multi-fading, better weather penetration, and more mobility than LP antennas. For CP antennas, the operating bandwidth is determined as a frequency band in which the bandwidth where the reflection coefficient *S*_11_ < −10 dB and that where axial ratio AR ≤ 3 dB overlap [23].

Various DRAs with circular polarization have been designed by exploiting different dielectric resonator (DR) geometries and feeding configurations. Feeding configuration is also important to vary the resonant frequency and the quality (Q) factor. The usual designs of the feeding configurations can be separated into two types which are single feed and dual or multiple feeds. Generally, the CP fields of a DRA can be excited by a single feed, such as by using a slot-coupling feed [24], coaxial probe feed [25], or microstrip-line feed [26]. The microstrip-line feed is the simplest feed, where the DR is directly placed on the transmission line which is printed on the substrate. It has been shown in [27] that the overlap distance between the DR and the microstrip-line feed controls the coupling strength and the specific mode that is excited by the transmission line. As the overlap distance is shorter than one quarter of a dielectric wavelength of the resonance frequency, the strongest coupling happens. The main limitation of the microstrip-line feed is that the feeding line is not isolated from the DR and may disturb the radiation performance of the DRA. Moreover, when the DR is placed directly on the top of a microstrip-line feed, an undesired air gap is formed between the DR and the substrate. A single-feed configuration suffers from a narrow AR bandwidth and severely restricts the applications of DRA. A dual or multiple feed configuration can also be used to excite CP waves. A dual or multiple feed configuration is based on the sequential phase (SP) rotation technique to create CP radiation with low cross-polarization and a higher AR bandwidth. This configuration requires a power divider to create CP radiation and the feed points should be orthogonal to each other with a 90° phase difference [28]. Typically, a CP DRA with dual or multiple feeds has a wide AR bandwidth but has a complicated design, bulky size, and high manufacturing cost as compared to single-feed CP techniques.

Review papers on CP DRAs exist in the literature [29,30,31]. However, the authors in [29] summarized their general thoughts about DRA, the advantages, and a comparison of the recent techniques used to obtain wideband CP DRAs. Whereas, in [30], the authors focused on the summarization of different techniques to generate a dual-band CP DRA only. Moreover, in [31], the authors summarized the basic concept of the CP mechanism for DRA in addition to the methods that can be used to obtain circular polarization in DRA. Previous review papers published in [29,30,31] summarized the methods of generating circular polarization in DRA and the advantages in general. However, a detailed comparison in terms of efficiency, modes, and applications using DRA are not covered by any other previous papers to the best of our knowledge. Therefore, our work not only includes a comprehensive review of different methods of producing circular polarization waves and the advantages, but additionally lists several applications using DRA and its important parameters. The paper is organized as follows: Section 2 explains the theory of dielectric resonator antenna. This section also explains the different methods of single- and dual-feed or multiple-feed DRAs to achieve CP. Moreover, a brief comparison in terms of resonant frequency, bandwidth, gain, efficiency, and mode is presented in Section 2. In Section 3, the performance of several multiple-input-multiple-output (MIMO) CP DRA is accessed in terms of the reflection coefficient, CP bandwidth, feed type, gain, and isolation. Section 4 comprehensively elaborates on a millimeter-wave CP DRA. Various applications of DRA are listed in Section 5. Lastly, the conclusion of this paper is presented in Section 6.

## 2. Dielectric Resonator Antenna

A dielectric resonator antenna is a microwave antenna that consists of a high dielectric material and Q factor, which is mounted on top of a ground plane, as shown in Figure 3. A dielectric resonator can be in various shapes such as rectangular, cylindrical, spherical, and hemispherical [32]. The rectangular DRA has several benefits over the cylindrical, spherical, and hemispherical DRA. A rectangular DRA offers two degrees of freedom, where the aspect ratio of height to width and depth to width directly influence the impedance bandwidth and the radiation Q factor [33].

Such as most resonant antennas, one of the significant properties of a DRA is the resonant frequency of the fundamental mode. Figure 4 shows a rectangular DRA fed by a slot aperture. The antenna consists of a rectangular DR with where width is *a*, length *d*, and height *b*. As it is characterized by three independent geometrical dimensions, *a*, *b*, and *d*, it offers more design flexibility as compared to the cylindrical DRA. Additionally, the cross-polarization level of a rectangular DRA is low compared to the cylindrical DRA.

TE modes are excited when the DRA is mounted on a ground plane. The resonant frequency of the fundamental mode, TE_111_, is calculated using the following equations [34].
(1)$$f_{0}=\frac{c}{2\pi \sqrt{\varepsilon_{r}}}\sqrt{k_{x}^{2}+k_{y}^{2}+k_{z}^{3}}$$
(2)$$k_{x}=\frac{\pi }{a}$$
(3)$$k_{z}=\frac{\pi }{2b}$$
(4)$$b=\frac{2}{k_{y}}\tanh\left(\frac{k_{y0}}{k_{y}}\right)$$
(5)$$k_{y0}=\sqrt{k_{x}^{2}+\;k_{z}^{3}}$$
where *ε_r_* is the relative dielectric constant of the resonator.

For cylindrical DRAs, the antenna consists of a cylindrical dielectric resonator with height *h*, radius *a*, and dielectric constant. The main advantages of the cylindrical DRA are the ease of fabrication and the ability to excite different modes. The resonant frequency of the modes can be calculated using Equations (6) and (7) [35].
(6)$$f_{TE_{npm}}=\frac{c}{2\pi \sqrt{\varepsilon_{r}\mu_{r}}}\sqrt{{\left(\frac{X_{np}}{a}\right)}^{2}+{\left(\frac{\left(2m+1\right)\;\pi }{2h}\right)}^{3}}$$
(7)$$f_{TM_{npm}}=\frac{c}{2\pi \sqrt{\varepsilon_{r}\mu_{r}}}\sqrt{{\left(\frac{X_{np}^{\prime }}{a}\right)}^{2}+{\left(\frac{\left(2m+1\right)\;\pi }{2h}\right)}^{3}}$$
where $X_{np}$ and $X_{np}^{\prime }$ are the roots of the Bessel functions of the first kind and of the relevant first-order derivatives, respectively.

Another important property of a DRA is its Q factor. The Q factor can be calculated analytically for canonical shapes, but for DRAs of arbitrary shape, the calculation of the Q factor relies on numerical methods or measurement. The Q factor of each mode can be calculated using the basis formulation derived by [36]. Considering only loss from radiation, the Q factor of each mode of the DRA is calculated using Equation (8).
(8)$$Q_{rad}=\frac{2\omega_{0}max\left\{W^{e},\;W^{m}\right\}}{P_{rad}}=\frac{2\omega_{0}max\left\{W_{vac}^{e}+W_{mat\;}^{e},\;W_{vac\;}^{m}\right\}}{P_{rad}}\;$$
where $W_{vac}^{e}$ and $W_{vac}^{m}$ are the stored electric and magnetic energy in free space, $W_{mat\;}^{e}$ is the stored electric energy in material, and $P_{rad}$ is the radiated power. $W_{vac}^{e}$, $W_{vac}^{m}$, $W_{mat}^{e}$, and $P_{rad}$ can be evaluated using the current and charge density within the dielectric object.

DRAs are an emerging antenna technology and a good option to fulfill the demands of being low profile and compact, while having a wideband characteristic. DRAs have small dissipation loss as they are made of a material with a high dielectric constant material and have no conducting parts. Therefore, DRAs can be highly efficient radiators. Moreover, DRA have small physical dimensions due to their high dielectric constant. Additionally, DRAs have wide bandwidth, high radiation efficiency, high gain, extra design flexibility, and simple excitation methods [37,38]. The CP DRA has gained much attention due to the combined features of both circular polarization and a dielectric resonator. The advantages and disadvantages of DRAs are summarized in Table 1.

In early years, DRAs were typically constructed of ceramic materials characterized by a high permittivity and high Q factor (between 20 and 2000). However, the fabrication of ceramic DRAs is quite challenging because the ceramic material is hard and difficult to machine [33]. A new approach is to use polymer-based materials such as plastic poly vinyl chloride (PVC), which make the fabrication process easier due to the natural softness, and results in a wide impedance bandwidth due to the very low permittivity of polymers [39]. One of the main advantages of DRAs is that various feeding techniques can be used to excite the radiating modes of a dielectric resonator.

### 2.1. Single-Feed CP DRA

The single-feed CP DRA has a simple configuration because no hybrid coupler is required to yield the 90° phase difference between the orthogonal degenerate modes [40]. A single-feed DRA achieves circular polarization by exciting two quasi-degenerate modes in the DR which are typically in phase quadrature and spatially orthogonal to each other. Usually, a single-feed CP DRA has a narrow AR bandwidth compared to the dual or multiple feeds type [41]. The AR bandwidth can be enhanced by a slight modification in the DR’s shape, by loading a metal strip onto the DR, and adjusting the feed structure. Metal strip [42], microstrip feedline [43,44], coaxial probe [45,46,47,48], and aperture coupling [49,50,51,52] methods are commonly used in single-feed antennas. The selection of the feed arrangement may depend on the antenna’s application. A microstrip feedline is very easy to fabricate because the feed is engraved on a similar substrate. However, excitation of the DRA using a microstrip feedline is very difficult, especially when the dielectric resonator has a low dielectric constant. When the dielectric constant of the DRA is equal to or more than 20, a high degree of coupling is realized because a high dielectric constant material will concentrate the field under the DRA. Thus, a high dielectric constant material is required at the bottom layer when the microstrip line is used to feed the structure [53]. Conversely, increasing substrate thickness will increase the spurious feed radiation and surface waves. However, this resulted in the bandwidth deterioration of the antenna and led to unwanted cross-polarized radiation.

A single-feed CP rectangular DRA has been designed in [42]. The proposed antenna is excited by using a H-shaped conformal metal strip, as shown in Figure 5. This proposed configuration does not require any multi-layering or any complex cutting of the DRA to obtain a wide impedance bandwidth of ~27.7% from 3.67 to 4.73 GHz and AR bandwidth of ~20% from 3.67 to 4.4 GHz. With a great gain of ~6.8 dBic, this antenna is suitable for worldwide interoperability for microwave access (WiMAX) and satellite applications. However, it is challenging to obtain a wide bandwidth and a good axial ratio because numerous modifications on a H-shaped feeding strip need to be performed. This will require extra time and work.

In addition, a coaxial probe can be used to feed a DRA. The connector’s inner conductor extends through the dielectric and is soldered to the radiating element, whereas the outer conductor is connected to the ground plane. In [45], two similar truncated corner liquid dielectric resonators (LDRs) are fed by a single coaxial probe for stimulating the essential TE_δ11_ mode of the LDR antenna, as presented in Figure 6. By changing the dimension of the truncated corner rectangular dielectric resonator and the probe’s position, a 90° phase difference between the orthogonal resonant modes is realized, resulting in CP radiation. The proposed antenna will obtain left-hand circular polarization (LHCP) if the ethyl acetate solution is inserted into the left side zone of the container and right-hand circular polarization (RHCP) if the solution is inserted into the right side zone of the container. A 35.6% operating bandwidth is recorded from 2.08 to 2.98 GHz, covering the broad 3 dB AR bandwidth of 16.3%, ranging from 2.31 to 2.72 GHz. However, the permittivity of the container will affect the impedance bandwidth and AR of the antenna. In addition, the antenna has a lower radiation efficiency due to the presence of the PIN diodes. Thus, a liquid with a suitable dielectric constant and low loss is required to increase the radiation efficiency.

In [46], a simple cylindrical-shaped DRA is proposed. The concept of PMC boundary approximation is used to generate a circular polarization wave in a dielectric resonator by changing the air-dielectric interface geometrical profile so that the E-fields change their route by disintegrating their components at dissimilar angles following the applied adjustment. The geometrical shape of a cylindrical dielectric resonator has been deformed periodically with the same angular factor (β = 45°) along the azimuthal direction, as shown in Figure 7. A coaxial probe is applied to stimulate the dielectric resonator fed through a small hole in the ground plane. The proposed DRA exhibits a 51.2% impedance bandwidth, ranging from 3.37 to 5.69 GHz and an 8.01% AR bandwidth from 4.2 to 4.55 GHz. The main advantage of this probe feed is it can be placed at any preferred location to match with its input impedance. The input impedance of the DRA can be tuned and the resonance frequency can be controlled by optimizing the length and position of the feeding probe. This feed type provides high coupling to the DR which, in turn, results in a high radiation efficiency. However, it has a narrow bandwidth of 2–5% and is hard to fabricate because a hole has to be drilled in the substrate, and the connector should protrude outside the ground plane. The dimensions of the drilled hole need to match the dimensions of the probe otherwise the effective dielectric constant of the resonator will be affected, thus the resonance frequency of the antenna will be shifted. 

In [48], a new converging fractal geometry is proposed for C-band applications. Fractal structures are created by using recursive procedures which produce large surface areas in limited space. Thus, the geometries and dimensions of fractal structures are essentially the deciding factor for the operation of resonant frequencies [54]. The self-similarity of fractal shapes is achieved by applying the infinite number of iterations. A fractal antenna offers size miniaturization by increasing the effective electrical length in the limited space. The perturbations introduced in the hemispherical DRA (HDRA) through the application of a fractal shape provide an enhanced impedance bandwidth because the removed DR reduces the Q factor with each step of iteration. However, it is a challenge to apply fractal geometry on HDRA because it offers a zero degree of freedom.

Aperture coupling is also one of the most commonly applied feeding techniques for CP DRAs. It is an indirect method, where the input signal is coupled to the radiating elements through the aperture (slot). In [49], a 50 Ω microstrip feedline is used and printed on the backside of the Rogers RO4535 substrate, as indicated in Figure 8. The proposed antenna operates in the TE_11,11_ higher-order mode from 10 to 13 GHz. The unequal slots are engraved on the copper ground plane with an angle of 45° concerning the microstrip feedline to excite two near-degenerate orthogonal modes of equal amplitude and a 90° phase difference for circular polarization generation. By joining a higher-order mode DRA, an outer dielectric layer and a cross-slot excitation, and broader impedance and CP bandwidths of ~21% and 9.5%, respectively, together with an improved gain of ~11 dBic, are achieved. However, this antenna suffers from a slight drop in the measured gain due to the presence of the uneliminated air-gap spots between the DRA and the dielectric coat.

A wideband CP DRA using a single feed can be obtained using parasitic vertical plates as proposed in [51]. By presenting four metallic plates around the dielectric resonator, a very wide AR bandwidth can be generated with the contribution from the cross-slot-fed DRA, the revolved metallic plates, and the interaction among the dielectric resonator and metallic plates. The AR bandwidth of the conventional DRA is successfully enhanced from ~10% to ~46.9% without increasing either the dimension or height of the antenna. The proposed antenna is fed with a microstrip-coupled cross-slot at the bottom, as shown in Figure 9. However, the position of the metallic plates (*dx*, *dy*) has an effect on the AR. The AR value will be degraded if the value of *dx* is 0 or negative. Thus, we need to maintain the positive value of *dx* for a good AR value.

In [52], a dual-band dual-CP DRA is proposed for unmanned aerial vehicle (UAV) applications. It consists of a metal strip with a rotated angle, a stacked rectangular dielectric resonator, and a substrate integrated waveguide cavity with a slot for feeding, as shown in Figure 10. The shared-aperture merged-structure design of DRA and a slot-dipole antenna are applied to realize circular polarization features. The CP waves are controlled by a phase compensation method with tuning processes. It is easier to obtain impedance matching by using this feeding technique than a contacting feed. However, the manufacturing process of stacked DRAs is complicated and needs to be carefully optimized in order to avoid any errors. A summary of some previous works on various single-feed circularly polarized DRAs are presented in Table 2. It can be concluded that a single feed has a simple geometry because no hybrid coupler is required to generate a 90° phase shift between the two orthogonal modes. CP waves can be realized by modifying the structure of the dielectric resonator to excite two orthogonal modes and the required phase shift. In addition, the dielectric permittivity *ε_r_* of the dielectric resonator and its shape will affect the performance of an antenna. Thus, by choosing a dielectric resonator with a high dielectric permittivity, the size of the DRA will be reduced. Moreover, each mode of a DRA has a unique internal and external field distribution. Therefore, different radiation characteristics can be realized by exciting different modes of a DRA.

### 2.2. Dual- or Multiple-Feed CP DRA

More than one feed antenna requires a power divider and a polarizer to excite CP radiation. The feed points need to be orthogonal to each other with a phase difference of 90° [55]. This type of feed generates TM_10_ and TM_01_ modes with an equal amplitude and are 90° out of phase. Each mode radiates individually and combines to yield CP. By using this technique, the AR bandwidth is improved. It can be as wide as the impedance bandwidth depending on the bandwidth properties of the phase shifter used, as this type of antenna is not dependent on the mode perturbing element. However, it suffers from a complex design, large size, and a high manufacturing cost compared to single-feed CP techniques.

In [28], a bowtie dielectric resonator is produced by two rotationally symmetric triangular ceramic elements. A vertical flat metal strip is attached to one of the triangular DRs and connected to the 50 Ω microstrip line by a probe through the ground plane. On the other hand, the second triangular DR is excited by cutting an L-shaped slot inside the ground plane, as shown in Figure 11. The lower CP band is formed by the slot mode and fundamental DR modes and the upper CP band is mainly formed by the slot mode and high-order dielectric resonator modes. The impedance bandwidth, AR bandwidth, and gain have been enhanced by using this sequentially rotated technique. The results indicate that the proposed antenna has wide overlapping working bands of 27.1% for the LHCP at the lower band and 12.8% for the RHCP at the upper band. Hence, this antenna is suitable for Wi-Fi applications, mobile-satellite services, radio navigation systems, and radiolocation equipment. However, this antenna requires high-precision fabricating technology and a low-loss feeding structure to enable it to work in a high frequency band.

A 2 × 2 sequentially rotated rectangular DRA array is proposed for 5.8 GHz applications in [56]. The dielectric resonators are coupled to the eye-shaped slots in order to generate circular polarization. Then, the array is fed by two interlaced ports and each port excites two radiating elements for the sake of size miniaturization, as shown in Figure 12. The proposed antenna has good gain and good port isolation with reduced size. However, this antenna exhibits a narrow bandwidth at ports 1 and 2 at a range of 5.769 to 5.913 GHz and 5.788 to 5.922 GHz, respectively. Some modification on the dielectric resonator geometry can be employed for bandwidth enhancement. Moreover, the number of the array elements can be increased to enhance the gain.

A polarization reconfigurable cylindrical DRA with tunable feed network is presented in [57]. The feed network comprises of Wilkinson power divider, two pairs of shifters and four pairs of PIN diodes, which generates two-way signals of equal amplitude and phase or quadrature phase to achieve LHCP, RHCP, or LP states. However, the measured impedance bandwidths are slightly different than the simulated one due to the influences in the dc biasing network and errors in the fabrication process. In [58], a dual CP DRA with a much wider AR bandwidth is offered for electromagnetic energy harvesting. As shown in Figure 13, a rectangular DR, composed of two microwave dielectric ceramics, is employed on top of a series feeding structure. The feeding structure consists of four intersected slots, a microstrip ring with series feeding lines, and two matching microstrip lines. The measured results indicate that the antenna has a 3 dB AR bandwidth of more than 43.5% over the operating band of 1.83–2.85 GHz and peak radiation efficiencies of about 90%. The DRA generates RHCP fields in the direction of broadside when port 1 is excited, and LHCP fields in the direction of broadside when port 2 is excited due to the symmetrical structure. Thus, this antenna is classified as a dual CP DRA. However, a slight difference between the simulation and measurement results is recorded due to the adhesive and air gap between the DR and the substrate. Moreover, errors during fabrication and measurement need to be avoided to achieve the best results. The summary of previous works on various dual or multiple feed DR and techniques to generate circular polarization is presented in Table 3. It can be concluded that the CP bandwidth increased and was almost the same as the impedance bandwidth depending on the properties of the phase shifter used. The feeds will stimulate the dual orthogonal modes for obtaining circular polarization.

The advantages and disadvantages of different feeding configurations are summarized in Table 4. Using a metal feeding strip and microstrip feeding is the easiest way and simplest method to feed the antenna. However, the bandwidth of the antenna is frequently affected and dependent on the thickness and dielectric permittivity of the substrate. For a coaxial probe feed, the soldering and drilling process during fabrication is difficult and requires extra care to make sure the conductor tracks on the printed circuit board (PCB) are not damaged. The antenna design will become complex if aperture coupling is used because it requires multilayer structures and the alignment is very important for good input matching [33]. To generate a CP DRA, sequential rotation of DR with a SP feeding network is a good option as it has a simple structural geometry; thus, it is easy to design and fabricate [59,60,61,62]. Through this method, a stable phase difference is produced at the ports of the feedline, resulting in the CP characteristics. Moreover, this technique allows the size of the antenna to be reduced.

## 3. Multiple-Input-Multiple-Output CP DRA

In recent years, multiple-input-multiple-output (MIMO) technologies have been in continuous development for modern telecommunication systems such as for wireless local area networks (WLANs), long-term evolution (LTE), and fifth-generation (5G) communications. In a practical communication system, the space for the MIMO antenna is generally limited. Moreover, the mutual coupling between antennas will degrade the efficiency and performance of the MIMO system. It also can affect the antenna’s radiation pattern. MIMO antennas designed through patch antennas tend to have lower efficiencies and insufficient isolation between unit antennas, rendering them unsuitable in most applications. Compact DRAs with high radiation efficiency will be much more favorable to combat the issue, observing no unwanted effects even in array configurations with better isolation. Numerous works on reducing mutual coupling between DRA elements have been reported, such as by using defected ground structures (DGS), electromagnetic band-gap (EBG) structure, metamaterials, and frequency selective surfaces (FSSs). DGS technique is applied in [63,64] to control the impedance transmission capacity and reduce mutual coupling. However, the proposed antennas are LP and would suffer in antenna misalignment, thus degrading the antenna performance.

In [65], an EBG structure was added between two CP DRAs to reduce the mutual coupling. The proposed antenna consists of a rectangular DR placed on top of the cross-ring slot, as shown in Figure 14. The two opposite corners of the DR are truncated at 45° and combined with a cross-ring coupling slot with an optimum ratio to obtain CP radiation with a wideband AR performance. The antenna is fed with a 50 Ω microstrip feedline for good impedance matching. A total of 25.9% and 18.7% measured impedance and AR bandwidths, respectively, are recorded from a 3.3 to 3.8 GHz frequency range for 5G applications. As this design has decoupling structures placed between the antenna elements, the system complexity will be increased.

In [66], an integrated four-element DR based on a MIMO polarization diversity antenna is demonstrated for vehicular communication. The proposed antenna comprises two conformal open-loop fed DRA elements (Ant. 1 and Ant. 2), and two microstrip-slot fed DRA elements (Ant. 3 and Ant. 4). As shown in Figure 15, Ant. 1 and Ant. 2 are located at the top of the substrate and placed diagonally. RHCP and LHCP waves are excited in Ant. 1 and Ant. 2, respectively, to establish polarization diversity. Similarly, Ant. 3 and Ant. 4 are positioned at the substrate’s bottom side and generate LHCP and RHCP waves, respectively. Thus, it will create polarization diversity and reduce the interference between the field components. The proposed antenna obtains a 500 MHz measured impedance bandwidth from 3.22 to 3.72 GHz, and a 200 MHz measured AR bandwidth from 3.34 to 3.54 GHz. However, to meet the needs of a small size MIMO antenna, the antenna will suffer from a large mutual coupling effect and a reduction in the antenna performances.

Metal vias can also be used to decrease the mutual coupling between the DRA elements. In [67], four metal strips on the lateral sides of each DRA are introduced, as shown in Figure 16. With that, the rotation direction of the coupled E-field in the passive DRA can be changed and made orthogonal to that of the active DRA. Owing to the polarization orthogonality, the isolation between the two DRAs is enhanced to 31 dB at 2.43 GHz without affecting the reflection coefficient, radiation pattern, and AR. The measured impedance bandwidth of the proposed antenna is approximately 9.3% from 2.36 to 2.59 GHz, and the measured AR bandwidth is 4.9%, ranging from 2.39 to 2.51 GHz. The length of the orthogonal rectangular slots should be unequal to generate the CP field. Otherwise, the antenna will radiate in linear polarization.

Another method that can lessen mutual coupling is orthogonally polarized [68,69]. In [68], the antenna elements are placed in orthogonal alignment to obtain an isolation greater than 15 dB between port 1 and port 2. In [69], two symmetric orthogonal feed networks are deployed to reduce mutual coupling between the ports. The recorded mutual coupling between port 1 and port 2 is well below 14 dB throughout the operating impedance band for WLAN applications. Table 5 summarizes some previous works on various MIMO CP DRAs and techniques to achieve low mutual coupling. Based on the table, the decoupling method is the best in enhancing the isolation of the antenna due to the orthogonal polarization caused by the metal strips on the DRA. This method is simple and does not require extra space. Moreover, the proposed design has circular polarization features with good MIMO characteristics.

## 4. Millimeter-Wave CP DRA

The rapid increase in the demand for mobile internet, the internet of things (IoT), and automated vehicles has motivated researchers to focus on highly efficient antennas, which exhibit wide bandwidth, high data rates, and good radiation characteristics to handle information exchange. Thus, millimeter-wave (mm-wave) bands have received significant consideration for their competency in avoiding interference with the overcrowded lower frequency spectrum. A 270 GHz bandwidth ranging from 30 to 300 GHz has been allocated for mm-wave communication systems. Various antennas such as patch, monopole, and dipole antennas have been proposed for mm-wave applications. However, these antennas suffered from a narrow impedance bandwidth and radiation efficiency problems because of lossy silicon substrates. Therefore, the DRA is a good option for mm-wave use owing to its high radiation efficiency, light weight, flexibility in resonator shape, and feeding scheme. At this mm-wave frequency band, DRAs will exhibit a wider impedance bandwidth and higher radiation efficiency, where the conductor losses of metallic patches are significant. Compared to the DRA operating in the microwave band, the sizes of the mm-wave DRAs are extremely small, and therefore, the practical realization issue becomes the main concern. 

Several studies have been carried out to overcome this drawback. However, most of them focused on LP designs [73,74,75] and very little attention has been paid to the mm-wave CP DRA [76,77,78,79,80]. In [76], four parasitic dielectric resonator elements are sequentially rotated with a 90° phase angle to achieve circular polarization, as shown in Figure 17. The proposed DRA is mounted on four layers of RT6010 substrates and fed by a printed gap waveguide (PGW). The function of the PGW is to eliminate the losses and dispersion of the substrate. The parasitic elements will cancel the cross-pol E-field component, thus, improving the CP performance. The proposed antenna has no aperture coupling and feed network system. Therefore, it leads to a reduction in the complexity of the feed and ohmic loss in mm-wave frequency. However, the proposed antenna will be thick as it is implemented by four layers of substrates and needs to be tightly bonded with metal screws.

In [77], a CP substrate-integrated CDRA has been proposed for 60 GHz applications. The DRA element has been isolated from the substrate using two circular arrays of holes and vias, as shown in Figure 18. A slotted circular patch is loaded on the DRA element at the top of the substrate to generate a CP field. The measured overlapping bandwidth between the reflection coefficient and AR has been increased by 15.9% from 55 to 64.5 GHz, with a peak measured gain of 11.43 dBic when the sequential rotation (SR) feeding method has been used for the 2 × 2 array. However, substrate-integrated antenna often leads to large leakage losses which depends on the separation of vias and holes.

One of the popular methods to enhance impedance bandwidth, AR bandwidth, and gain of the proposed antenna is by integrating a SP feeding network into the antenna design [78]. The DRA using a SP feeding network is proposed in for the mm-wave application at 30 GHz. As depicted in Figure 19, the proposed antenna consists of a 2 × 2 array of a flower-shaped DR placed on a ground plane. The cross-slots and SP feeding network are etched on the top and bottom sides of the Rogers RO3006 substrate, respectively. The array antenna is fed using a parallel feeding network, where each DR is fed sequentially, with a 90° phase difference between them in a SR manner. This arrangement is responsible for CP enhancement. Table 6 summarizes some former works on various mm-wave CP DRAs. The designs in [76,77] are quite complicated because they have multiple layers of substrate. In [78], the complex feeding structure of an antenna array resulted in ohmic and surface wave losses which tend to decrease the overall efficiency of the antenna. Moreover, an array antenna is difficult to design, especially at mm-wave frequencies.

## 5. Applications of DRA

There are many applications for DRA over MPA, which make it more appealing, and they are discussed below.

### 5.1. RF Energy Harvesting

RF energy harvesting refers to the utilization of the ubiquitous RF energy transmitted by different wireless systems to feed electronic devices with a low-power consumption remotely. It is widely adopted to replace conventional batteries. Recent growth in wireless technology has led to an enlargement of the existing RF energy in different frequency bands. Thus, a broadband antenna is required to capture the ambient RF energy over this wide frequency range. Moreover, the proposed antenna should also have a high radiation efficiency and high gain to maximize the received RF power from the ambient RF sources. In previous works, numerous planar wideband antennas have been developed to collect RF energy from several frequency bands [81,82,83,84]. However, most of them are unable to obtain the satisfactory gain. Hence, high input power is required to enhance the performance. Given that the ambient RF radiations are weak, unstable, and span over a wide range of frequencies, antennas with a wide impedance bandwidth, high gain, and high radiation efficiency are essential to collect a huge amount of power from the low-density environment. DRAs are proposed in [58,85,86,87,88] to overcome these problems. Compared to the planar antennas, DRAs offer a wide bandwidth, high gain, high radiation efficiency, low conductor loss, and compatibility to different feeding mechanisms.

A stacked rectangular DRA for RF energy harvesting is proposed in [85]. The DR is made from soda-lime glass. A slot feeding method is used to feed the antenna as it gives a better impedance matching, low spurious radiation, and better polarization purity owing to the indirect electromagnetic interaction between the DRA and the feed line. The DR is stacked at seven layers and a thin air gap is added to improve the gain to 6.69 dB. In the air gap technique, air is used as the substrate in between the ground and DR. Due to the use of air as a substrate radiation efficiency, the impedance bandwidth is enhanced, and the dielectric loss is reduced. However, the impedance bandwidth is limited to 18.8% only.

In [86], a broadband high-gain hybrid DRA is proposed, and a 120% impedance bandwidth is recorded from 1.67 to 6.7 GHz with the maximum gain value of 9.9 dBi. The proposed antenna consists of a rectangular DRA backed by a rectangular slot in the ground plane, as shown in Figure 20. The proposed antenna is excited by using an inverted T-shaped feed line which consists of three arms of different lengths. This feeding mechanism effectively couples the energy for a broad range of frequencies between the feed line and the radiating elements. In addition, a metallic reflector is placed below the antenna for gain enhancement.

Circularly polarized DRAs for RF energy harvesting are proposed in [68,87,88]. Different shaped geometry has been considered for bandwidth and gain improvement [87,88]. For instance, in [87], a one-fourth of the cylindrical DRA is proposed for bandwidth and gain improvement. A dumbbell-shaped slot is etched on the ground plane and placed on the other side of the substrate to achieve the resonance within the desired operating frequencies. In order to obtain circular polarization, two metallic strips V-shaped are etched on the upper edge of the DR. The proposed antenna can capture electromagnetic signals from all directions because it radiates in an omnidirectional direction pattern. Therefore, this makes it suitable for wireless energy harvesting applications in a smart city. In [68], the proposed antenna consists of a 90° twisted quarter sectored CDRA, a metallic plate, and a shunt-diode rectifier circuit. This antenna offers a high gain of 7.02 dBc. However, it suffers from a narrow bandwidth between 5.59 to 5.88 GHz. In addition, this antenna is complicated because the authors need to design and optimize the shunt-diode rectifier for rectification purposes. Table 7 summarized a comparison between various DRAs for RF energy harvesting. It can be concluded that by introducing a thin air gap between the DR and the ground plane, a reasonable bandwidth is achieved, and the gain is improved. Moreover, a thin air gap is easier and more practical to apply compared to a wide air gap. 

### 5.2. Radio Frequency Identification (RFID)

RFID is a revolutionary application of automatic identification and data capture technology that is contactless and does not require line of sight. It is used to identify, localize, and track an item or object provided with an RFID tag or transponder by the means of radio EM waves. It can be embedded in all kinds of consumer products and scanned several meters away, revealing information about the product and its manufacturer. Small size and low-cost RFID tags are crucially important for applications of the Internet of Small Things (IoST). There are several methods that can be used to reduce the size of RFID tags such as meandered antennas, fractal dipole antennas, and dielectric resonator antennas.

In [89,90], high-permittivity ceramic resonators are used for RFID tags, which are intended to provide size miniaturization and long-range reading distances. Moreover, ceramic materials can consistently operate under severe conditions such as extreme temperatures and chemically violent environments. In [89], a 22.9 m reading distance is obtained for the RFID tag with *ε_r_* = 100. In [91], a polarization twist tag is proposed. The twist tag consists of two dielectric resonators tilted by ±45° and spaced by λ/4 for orthogonal polarizations. Figure 21 shows the operation of this system. The DR tag will reflect an equal-handed CP wave. However, another reflector is required to reflect the opposite-handed wave for a clean resonance received by the co-polarization network of the reader with less degradations due to lens reflections.

In [92], a novel ultraminiature CP antenna inspired by crossed split-ring resonators (SRRs) is proposed. The proposed antenna consists of a F4BM top substrate board, metallic screws, and a metallic ground layer. Two crossed dual-capacitance ring resonators (C-DRRs), which are equivalent to two crossed magnetic dipoles, are arranged perpendicularly and excited simultaneously with an SMA connector, as shown in Figure 22a. As can be seen in Figure 22b, the measured −10 dB bandwidth is from 900 to 928 MHz which is 3.1%, whereas the simulated frequency range is from 901 to 928 MHz. The proposed antenna will be a good option for RFID and low frequency applications due to its ultracompact size, low cost, high radiation efficiency, and wide CP beamwidth. A PIN diode and varactor are used in this design to achieve polarization and frequency tuning. Undoubtedly, PIN diodes and varactors need an additional bias circuit.

Chipless passive tag technology has been applied to RFID and wireless sensor approaches which do not rely on microchips to form a backscatter signal. Recently, a DR has been proposed as a tag in a temperature sensor system. In this system, the energy is coupled by the reader into the DR by sending a signal at a resonant mode frequency and the DR will transmit back the signal effectively within the frequencies bands. However, a single or flat cylindrical DR tag exhibits low radar cross section (RCS), which extremely limits the reading range. Thus, a new method to boost the RCS is introduced by combining the DR tag and the spherical lens [93,94]. The focused incident wave is well coupled to the two lowest order resonant modes when the DR is placed along the focal area behind the lens. Different sizes of the DRs may be employed to create tag spectral signatures which can be discriminated without uncertainty in a multiple-tag situation. These chipless tags are suitable for indoor self-localization systems. However, the frequency-dependent and angle-dependent response of the DR tag response may affect the performance of the system, which then results in localization accuracy. Table 8 summarizes a comparison between various DRAs for RFID. 

### 5.3. Radar

Radar is a detection system that uses radio waves to measure distances, scattering parameters, or angles velocities. It is widely used in autonomous driving, security, defense, meteorology, and many more applications. Radar should have a high resolution, high efficiency, low false alarm rate, and small sensor dimensions in order to cope with various conditions depending on its application. 

The authors in [95] proposed a rectangular DRA planar array with a broad bandwidth of 50% and a high gain of 22 dBi for unmanned aerial system radar applications. The 8 × 8 array is designed using a stripline feeding network with two different substrates to improve shielding of a multi-stage feeding network. A planar array also being used in [96]. The proposed rectangular DRA obtains an impedance bandwidth of 6.98% centered at 24.09 GHz, with the antenna gains varying from 7 to 9.5 dBi. However, the spacing between the antenna elements should be proper and in exact phase to form a good radiation beam and enhance the directivity of the system. In addition, the feeding network of an array requires a large area to occupy multiple segments of transmission line. This will cause the antenna size to be larger. Moreover, the proposed antennas in [95,96] radiate in linear polarization. For an antenna with linear polarization, the electromagnetic waves broadcast on a single plane are either vertical or horizontal. If the signal is reflected inaccurately, interference will occur and the signal will lose its strength.

For conventional radar systems, it is tough to deal with varied scenes and to accurately identify and classify targets. Therefore, the radar sensor has to transmit and receive multiple polarizations that are affected differently by the observed scene. In [97], a four-element cylindrical DRA array with a simple dielectric fixing slab and glue is proposed for 24 GHz automotive radar, as shown in Figure 23. A conventional one-to-four microstrip power divider feeds the proposed antenna. Unevenness and air gaps need to be avoided by applying necessary and unnecessary glue during fabrication and assembly processes to mitigate the possible problem of glue spreading.

In [98], two antennas, element and array antennas, have been fabricated and investigated. The proposed antennas employ a quadrature hybrid structure to excite two perpendicular HEM modes for achieving CP radiation. The hybrid structure is integrated with a cylindrical DR, as shown in Figure 24. Based on the simulated and measured results, these antennas are good candidates for X-band radar applications such as weather monitoring and vehicle speed detection. However, there are certain conditions that need to be satisfied in order to generate the appropriate CP wave. First, the feeding points should be placed perpendicularly to realize the two perpendicular HEM modes. Next, equal amplitude modes must be generated between the two paths. After that, a 90° phase difference between output ports should be achieved by ensuring a quarter wavelength difference between the two paths. Lastly, a 50 Ω input impedance is crucial to get good matching. Table 9 summarizes a comparison between various DRAs for radar applications. 

### 5.4. Biomedical Applications

The advantages of DRA also make it a suitable candidate for biomedical applications [99,100,101,102,103]. Various biomedical applications have different requirements in terms of data rate, transmitter power, and link distance in order to ensure that discriminative information is delivered within a satisfactory timeframe for the most serious situations. In [99], a new coil setup for dual-nuclei imaging has been proposed by combining an annular DR filled with a high permittivity material for phosphorous and a traveling-wave antenna for ^1^H scout imaging and shimming. However, the DR has a lower sensitivity which is only half that of an equivalently sized birdcage on the phosphorous channel. A dual-polarized (DP) omnidirectional hemisphere DRA for a wireless capsule endoscope system (WCE) is proposed in [100]. The antenna was excited in TM_01δ_ and TE_01δ_ modes to radiate a DP wave by a feeding probe and four arc microstrip line, respectively. The omnidirectional DP wave are realized when the TM_01δ_ and TE_01δ_ modes are equal in amplitude and phase. The hemispherical shape of the DR is easy to be conformal with the end of the capsule. The proposed antenna has a gain between −13.5 and −15dBi, with an impedance bandwidth from 2.4 to 2.48 GHz. However, the measured reflection coefficient is slightly worse than the simulated one, mainly owing to a machining error.

A rectangular DRA operating at 2.45 GHz is proposed in [101] as an implantable antenna with no metallic losses, bio-compatibility, and a varied implant depth performance. The DRA are made up of biocompatible ceramics with *ε_r_* = 80. The implant DRA is mounted on a PVC substrate and fed by a coplanar waveguide. It is concluded that the reflection coefficient degraded as the depth of the implant antenna increases but still has an acceptable performance for the defined ‘depth window’. Moreover, the proposed antenna has acceptable radiation limits for the specific absorption rate (SAR) value. In [102], a singly-fed CP wearable DRA for off-body communications has been proposed. The antenna has been excited using a H-shaped conformal metal strip, as shown in Figure 25. This antenna radiates in circular polarization with an impedance bandwidth of 20.7% from 6.95 to 8.68 GHz. Moreover, this antenna recorded a low SAR value. However, the lossy and high dielectric constant characteristics of human body may affect the frequency and reduce the efficiency of the antenna. Further, the antenna designer should be aware of the antenna’s impact on the human body.

DRA can also be used for tumor detection [103]. The proposed antenna consists of a cylindrical DR fed by a microstrip line and operated from 2.2 to 2.6 GHz. Circular polarization is realized by aperture coupling using the cross slot. In this work, the breast phantom is placed between two cylindrical CP DRAs for obtaining the simulation results, as shown in Figure 26b. The simulation was carried out for various tumor sizes from 1 to 8 mm in order to test the sensitivity of the system. As the size of the tumor increases, the overall permittivity of the tissue increases and disturbs the phase of orthogonal components of currents; therefore, AR increases from 3.5 dB to 4.8 dB. Table 10 summarizes a comparison between various DRAs for biomedical applications.

### 5.5. Vehicular Applications

DRAs are also used in vehicular or unmanned aerial vehicles (UAVs) applications [52,66,104,105,106]. The UAVs are frequently called drones, flying cars, aerial vehicles, and other names. The UAVs can fly autonomously or can be operated by human pilots [107]. The tremendous growth of UAVs is due to their high aerial mobility, advanced battery technology, rotors, global positioning system (GPS), cameras, sensors, low cost, fuel efficiency, and a broad range of applications [108,109]. The UAVs provide new potential for business in civil and non-civil applications such as agriculture, parcel delivery, aerial mapping, wildlife conservation, and surveillance [110,111]. As the applications for the vehicles are growing rapidly day by day, lightweight, compact size, high-gain, wide bandwidth and high-efficiency antennas are in high demand to enhance the capacity of wireless systems, reduce the overall system loss, and save the energy [112,113]. Moreover, a fine-range resolution and accurate navigation are also necessary to ensure efficient and safe operation even in strong multipath environments and intended interference [114]. To solve the above issues, DRAs are used as they offer wide bandwidth, high gain, high efficiency, and low losses that make them excellent candidates for vehicular applications.

In [104], a multiband DRA is designed for LTE automotive application. The proposed DRA is mounted and measured on the vehicle rooftop, as shown in Figure 27. Two DRAs are placed within a small area to reconfigure their radiation patterns on each frequency band in order to improve both the quality and reliability of the wireless link. As the vehicle’s body is curved, the antenna should be flexible to withstand various bending conditions while still maintaining its normal performance under these deformations.

In [105], a low-frequency DRA integrated with a high-frequency dielectric lens antenna (DLA) is proposed for dual-frequency vehicle communication. Both of the antennas share a single dielectric body, thus making the structure compact. The DLA is fed by a second embedded cylindrical DRA which is intended to reduce the conduction loss in the high band while maintaining a compact dimension. The DRA is excited by a vertical conducting adhesive strip on its sidewall. The dual-frequency antenna can provide wide impedance bandwidths and features a high degree of flexibility without the limitation of the frequency ratio. Good impedance matching is observed in the two bands when a larger ground plane is used. However, increasing the size of the ground plane will reduce the boresight antenna gain as the low-band antenna 3-dB beamwidth is widened in the H-plane and introduces many ripples in the E-plane. Both of the antennas in [104,105] operated in LP.

A radiation pattern of an antenna is one of the important parameters because it demonstrates the direction of the radiated energy distributed by the antenna into space. In the past few years, several conical beam antennas have been proposed for various applications owing to their omnidirectional radiation in the azimuth plane and directional radiation in the elevation plane. Specifically, in drone and vehicular communications applications, the transmitter or receiver needs to operate at a specific elevation [106]. Therefore, a conical beam radiation pattern is required to ensure that the radiation from the antenna fully covers a 360° latitude. Moreover, a conical-beam radiation-pattern antenna with a high gain and wide beam coverage is necessary to cope with the relevant movement between the antenna and the satellite [115]. To produce a wideband conical beam radiation pattern, several methods have been implemented. The most common method is by using a cylindrical DRA. This is because a cylindrical-shaped DRA is easy to design compared to the complex geometries such as an equilateral triangle. In [116], a DRA with conical radiation patterns is presented for vehicle-mounted surveillance equipment, as shown in Figure 28. The proposed antenna consists of a cylindrical DR and a loaded annular column. It is then placed on a metal disk and fed by a coaxial probe. With this configuration, four conical radiation pattern modes are excited and merged, providing an impedance bandwidth of 56% from 3.14 to 5.56 GHz. However, this antenna radiates in linear polarization. 

Very few CP DRAs for vehicular applications or unmanned aerial vehicle (UAV) have been reported in the literature so far [52,66,117]. In [52], the shared-aperture merged-structure design of a DRA and slot-dipole antenna is applied to realize circular polarization features. The CP waves are controlled by a phase compensation method with tuning processes. It is easier to obtain impedance matching by using this feeding technique than a contacting feed. However, the manufacturing process of a stacked DRA is complicated and needs to be carefully optimized in order to avoid any errors. In [117], a wideband CP is realized from the orthogonally-placed conformal strips, off-centered circular slot, and curved path in the feedline. However, a slightly different outcome in the results occurred due to the presence of air gaps and errors during the fabrication process. Table 11 summarizes a comparison between various DRAs for vehicular applications.

### 5.6. Solar Cells

The requirement to get power from renewable resources is becoming more challenging owing to the current energy and environmental challenges. The integration of antennas with solar-cell panels offer compact integrated platforms to employ the electromagnetic spectrum in both microwave and optical regimes. Thus, various methods have been implemented to develop antennas with high performance and full compatibility with solar-cell panels. The most common method is to mount an antenna on the bottom side of the solar cells [118]. This method is simple and not affect the solar cells. However, this method has limited applications especially for broadside antennas. By integrating the antenna on top of solar cells, the required coverage will be improved for many applications, but it may reduce solar cell efficiency due to the shadowing effects of the antenna [119]. 

In order to reduce the shadowing effects, transparent conductive oxides were used [120,121]. However, transparent conductive oxide has a sheet resistance greater than 5 Ω/sq with a high optical transmittance and the sheet resistance reduces the radiation efficiency of the antenna. Meshed metallic structures [122] also can be used to improve shadowing effects, as shown in Figure 29. A metal mesh can overcome the drawbacks of conductive films. The open areas guarantee stable optical transparency over the visible light spectrum and the conductive mesh offers a higher sheet conductivity for better radiation efficiency. However, integrating metal-meshed antennas with solar cells might reduce the realized gain and transmission loss, especially at high frequency, due to the loss nature of solar cells and the anisotropy triggered by parallel electrodes on solar cells.

To overcome this limitation, transparent dielectrics can be used as resonator antennas and can be mounted on top of solar-cell panels. This method can enhance the performance of the antennas with less shadowing impacts. However, a transparent dielectric usually has a low permittivity, which might affect the antenna design. In [123], a transparent rectangular DRA is integrated on top of a silicon solar cell with glass protection. Horizontal metallic strips are attached on the sidewalls to achieve radiation modes with various farfield and impedance properties, as shown in Figure 30. The metallization on the sidewalls of the DR should be minimized to increase the transparency of the antenna in all directions and minimize the shadowing effects. Table 12 summarizes a comparison between transparent antennas for solar cells.

## 6. Conclusions

This review article is mainly focused on the current development of CP DRAs and its applications. In recent years, tremendous efforts have been made on investigating the LP wideband DRAs and several methods have been proposed. However, there are not many studies about CP DRAs, especially for radar and UAV applications. Different methods in achieving CP DRAs that are emphasized in this article are intended to bring ideas for future research to develop high-performance methods for generating circular polarization waves. This article indicates that CP antennas are the best option for combating the drawbacks of LP systems such as polarization mismatch, fading effects, faraday rotation effects, and inclement weather conditions. However, most of the CP antennas suffer from a narrow bandwidth. A multilayer or stacked DRA with different sizes and dielectric materials is often used to enhance the bandwidth of the antenna. However, this method will increase the antenna size and is not suitable for several applications. Another method is by using special-shaped DRAs, but these DRAs may be difficult to obtain on the market. Several feeding configurations can also be used to obtain a wideband DRA such as by using a SP feeding network with a sequential rotation DR. This method also manages to reduce the size of a conventional antenna. As most of the applications require a compact size, high efficiency, and low loss, various shapes of DRAs are proposed and some of the design approaches are discussed. A dielectric resonator element has a low metallization and can increase the gain and radiation efficiency of the antenna while maintaining a circular polarization response with a wider axial ratio bandwidth. Additionally, a good impedance bandwidth and gain can be realized by introducing a thin air gap between the DR and the ground plane. Moreover, a thin air gap is easier and more practical to apply compared to a wide air gap. However, the impedance bandwidth is limited to 18.8% only. From our point of view, it can be concluded that an array DRA with a cross slot in the ground plane is the best method to generate a CP DRA. This design is simple and easy to obtain orthogonal modes by adjusting the length of the slots. Moreover, an array antenna exhibits a high signal strength and high directivity. Additionally, the feeding network also plays an important role in ensuring good antenna performances. This is because the feeding network will control and generate a phase difference to enable the CP wave and has the benefit of improving impedance and axial ratio performances. Thus, designing a suitable feeding network is important for obtaining a stable phase difference. As mentioned earlier, numerous works on ceramic DRAs have been performed at present. However, the main challenge for implementing ceramic DRAs is the hardness of the materials, which makes it difficult to fabricate. Therefore, new materials which are soft and flexible need to be proposed to replace the traditional ceramic materials such as by using polymer-based materials, water, etc. The information given in this review paper is useful for researchers who are working on CP DRAs.

## Figures and Tables

**Figure 1 micromachines-13-02178-f001:**
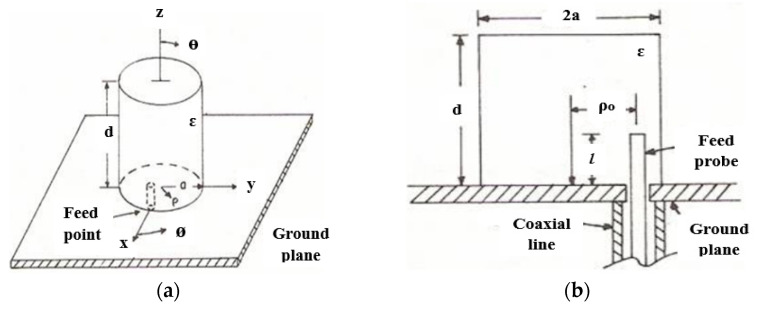
Cylindrical DRA: (**a**) full geometry; (**b**) feed configuration [15].

**Figure 2 micromachines-13-02178-f002:**
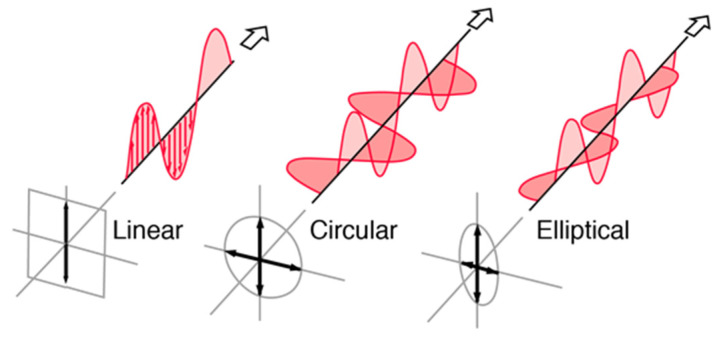
Types of polarization [17].

**Figure 3 micromachines-13-02178-f003:**
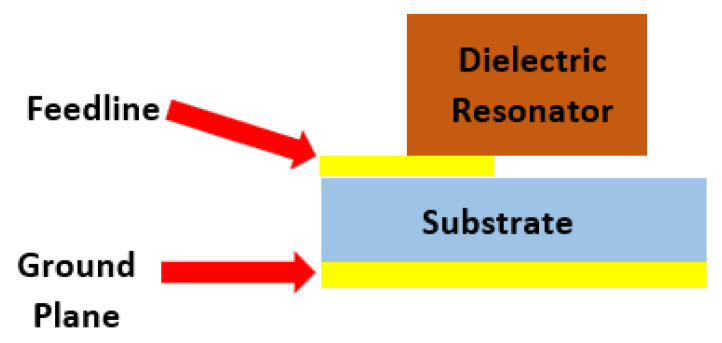
Typical configuration of DRA.

**Figure 4 micromachines-13-02178-f004:**
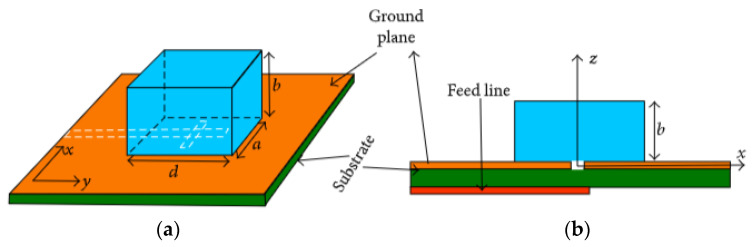
Rectangular DRA: (**a**) 3D view; (**b**) side view [33].

**Figure 5 micromachines-13-02178-f005:**
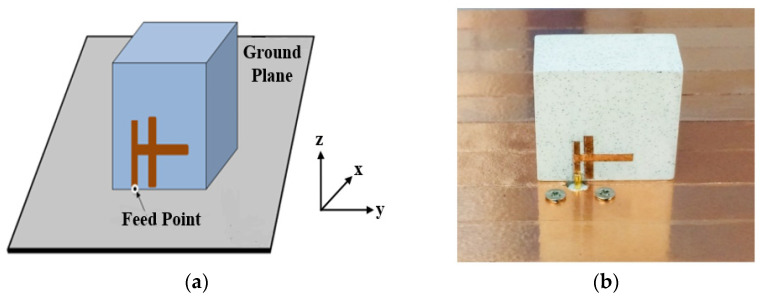
A singly-fed rectangular DRA: (**a**) proposed configuration; (**b**) fabricated [42].

**Figure 6 micromachines-13-02178-f006:**
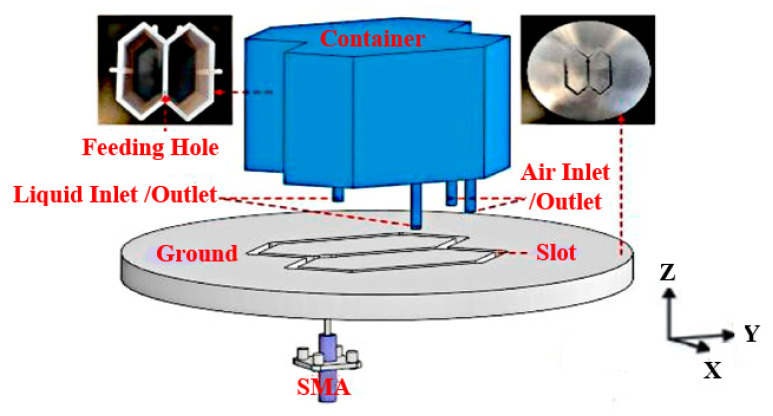
Proposed CP reconfigurable LDRA [45].

**Figure 7 micromachines-13-02178-f007:**
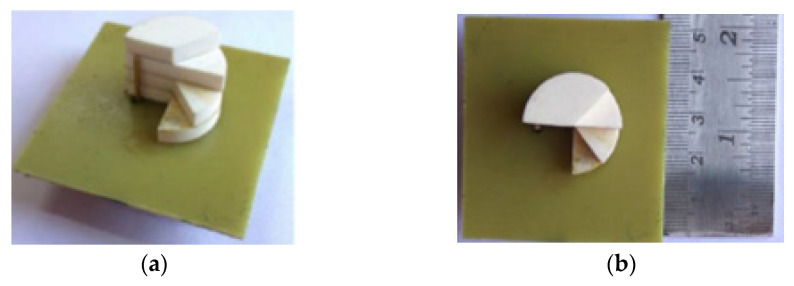
Fabricated DRA: (**a**) 3D view; (**b**) top view [46].

**Figure 8 micromachines-13-02178-f008:**
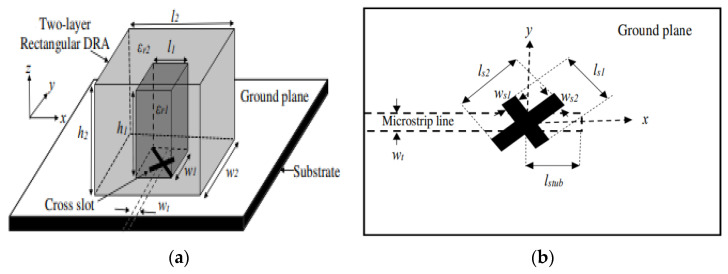
Proposed rectangular DRA: (**a**) 3D view; (**b**) top view of the microstrip feedline [49].

**Figure 9 micromachines-13-02178-f009:**
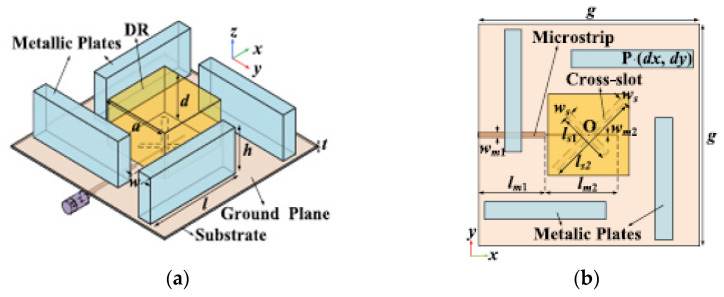
Proposed cross-slot-fed CP DRA: (**a**) perspective view; (**b**) top view [51].

**Figure 10 micromachines-13-02178-f010:**
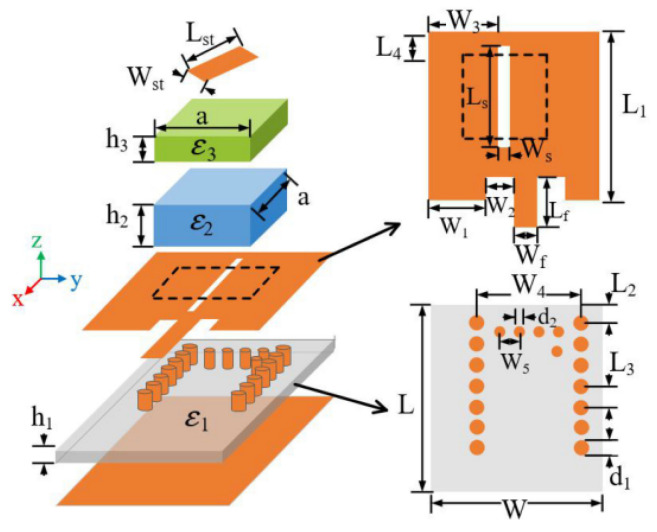
Overall configuration of a dual-band dual-CP DRA [52].

**Figure 11 micromachines-13-02178-f011:**
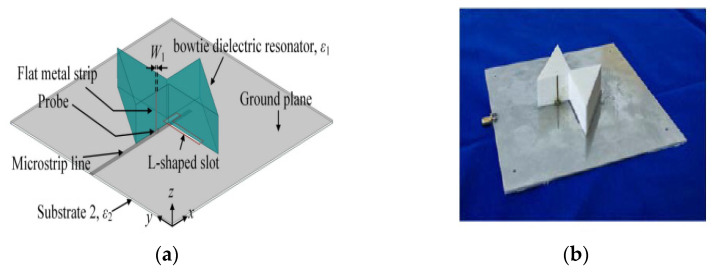
Proposed bowtie DRA: (**a**) 3D view; (**b**) fabricated DRA [28].

**Figure 12 micromachines-13-02178-f012:**
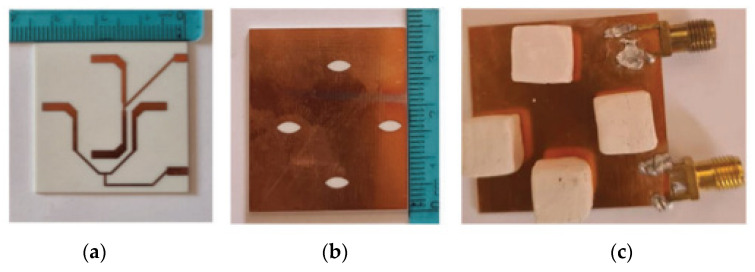
Fabricated RDRA array: (**a**) top plane; (**b**) bottom plane with eye slots; (**c**) bottom plane with dielectric resonators mounted over the eye slots [56].

**Figure 13 micromachines-13-02178-f013:**
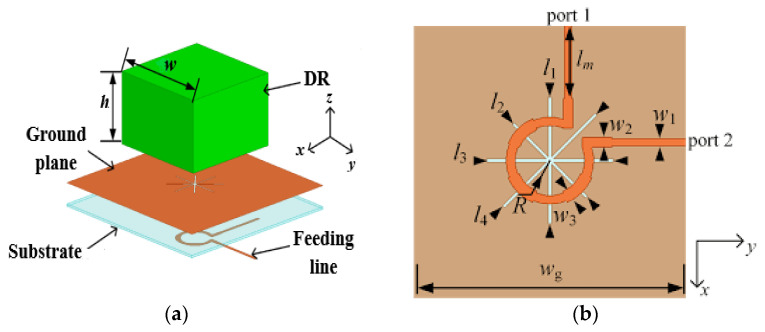
Proposed broadband dual CP DRA: (**a**) 3D view; (**b**) feeding network [58].

**Figure 14 micromachines-13-02178-f014:**
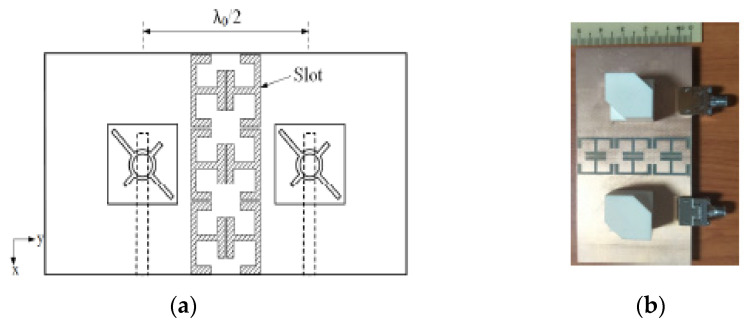
Proposed MIMO DRA: (**a**) front view; (**b**) fabricated antenna [65].

**Figure 15 micromachines-13-02178-f015:**
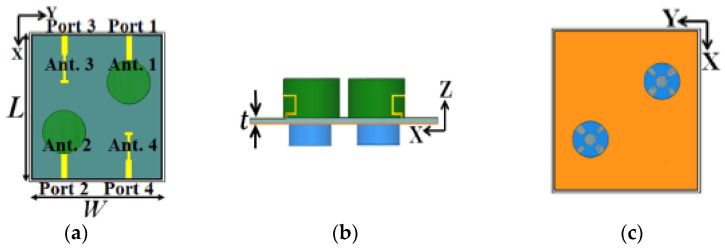
Proposed four-element MIMO DRA: (**a**) front view; (**b**) side view; (**c**) back view [66].

**Figure 16 micromachines-13-02178-f016:**
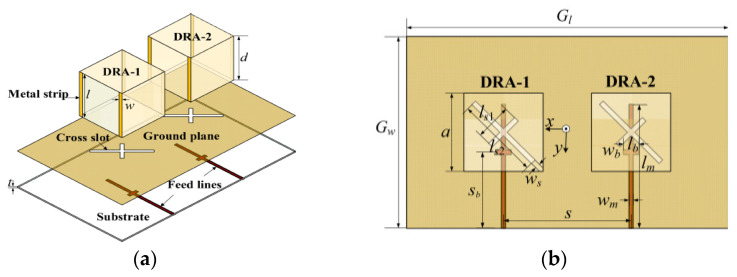
Configuration of the proposed MIMO DRA: (**a**) perspective view; (**b**) top view [67].

**Figure 17 micromachines-13-02178-f017:**
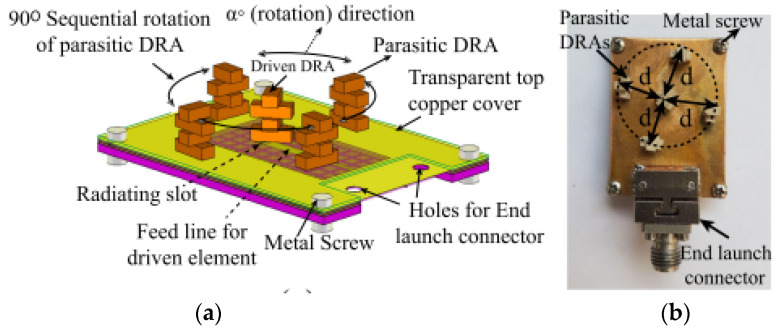
Parasitic DRA: (**a**) 3D view; (**b**) fabricated antenna [76].

**Figure 18 micromachines-13-02178-f018:**
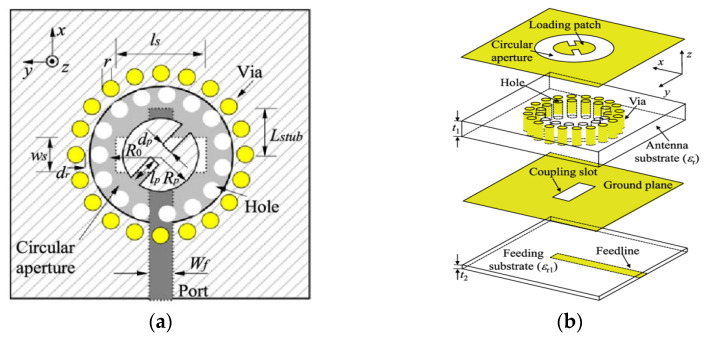
Geometry of CP substrate-integrated CDRA: (**a**) top view; (**b**) perspective view [77].

**Figure 19 micromachines-13-02178-f019:**
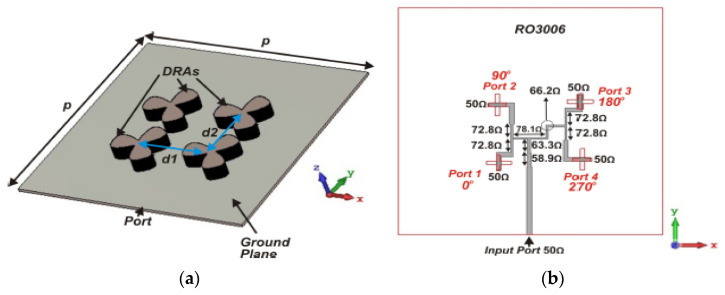
Proposed DRA: (**a**) perspective view; (**b**) bottom feed network [78].

**Figure 20 micromachines-13-02178-f020:**
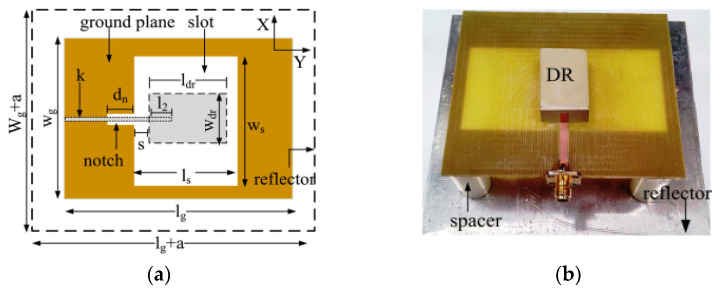
High gain DRA: (**a**) bottom view; (**b**) fabricated prototype [86].

**Figure 21 micromachines-13-02178-f021:**
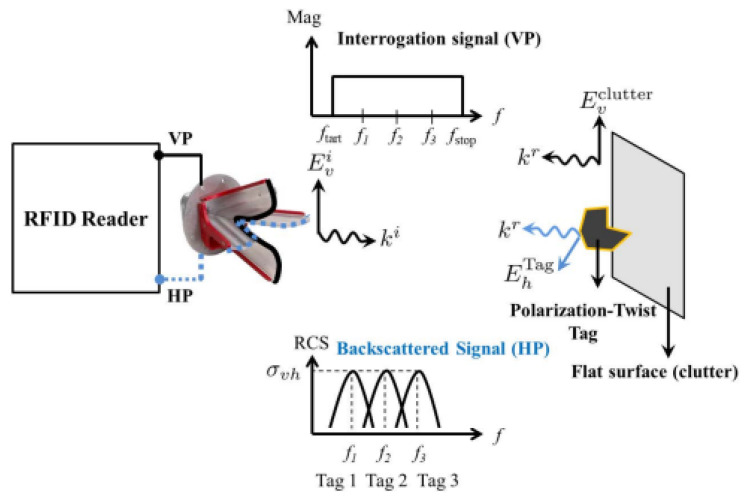
Operation principle of the proposed antenna [91].

**Figure 22 micromachines-13-02178-f022:**
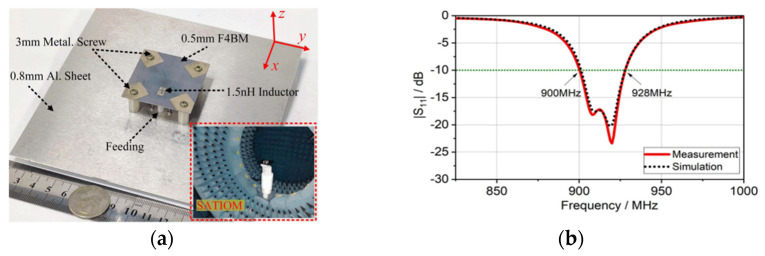
CP antenna with C-DRRs: (**a**) fabricated prototype; (**b**) reflection coefficient [92].

**Figure 23 micromachines-13-02178-f023:**
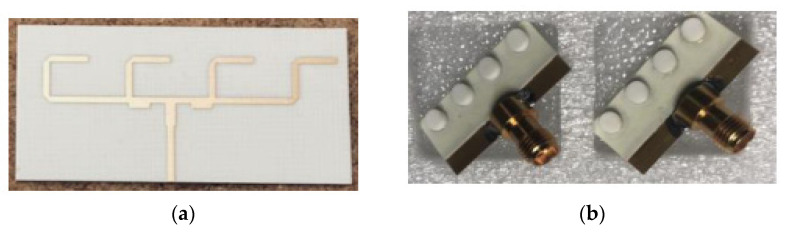
Fabricated cylindrical DRA array: (**a**) feed network; (**b**) array [97].

**Figure 24 micromachines-13-02178-f024:**
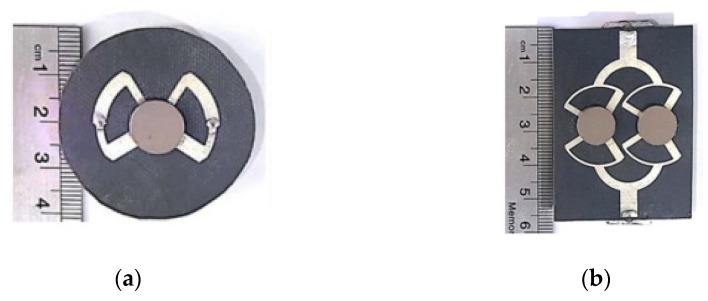
Fabricated prototypes: (**a**) element antenna; (**b**) array [98].

**Figure 25 micromachines-13-02178-f025:**
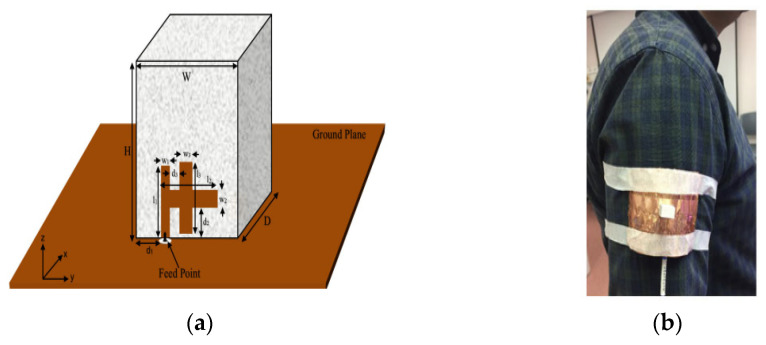
Wearable DRA: (**a**) proposed configuration; (**b**) antenna on a human body [102].

**Figure 26 micromachines-13-02178-f026:**
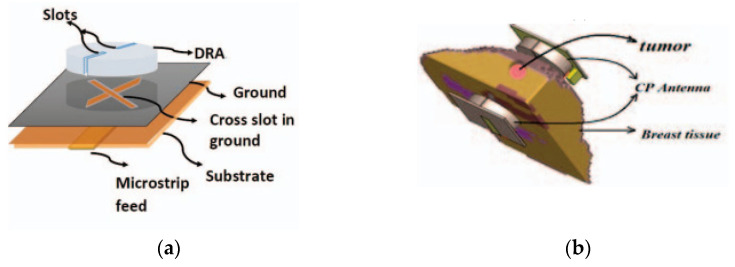
CP DRA: (**a**) overall configuration; (**b**) setup of the detection system [103].

**Figure 27 micromachines-13-02178-f027:**
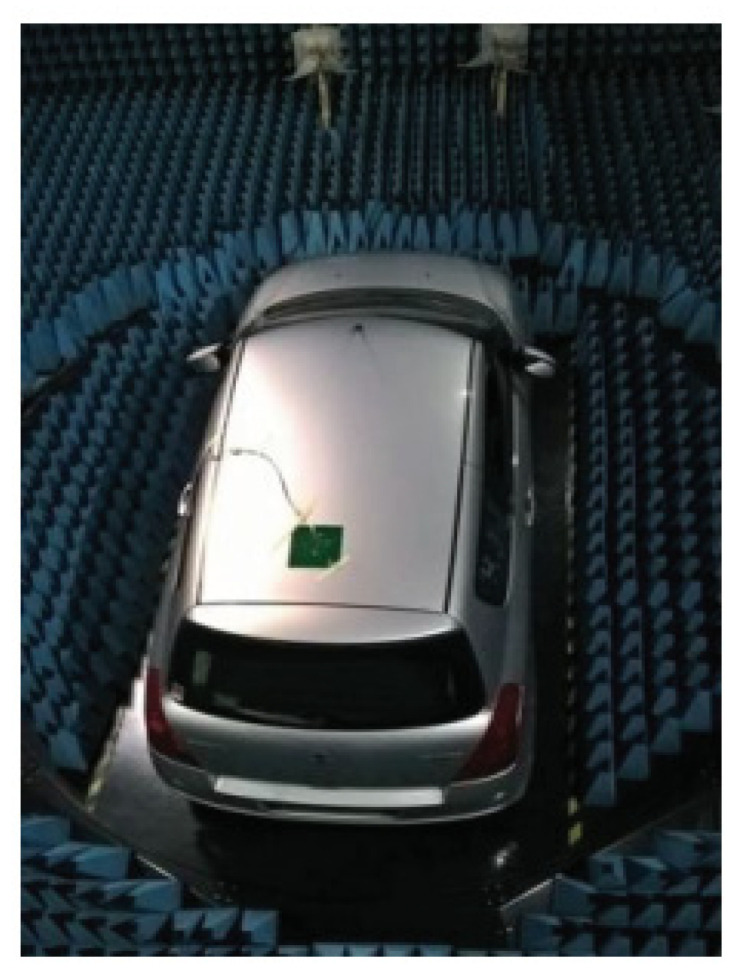
Proposed antenna mounted on vehicle’s rooftop [104].

**Figure 28 micromachines-13-02178-f028:**
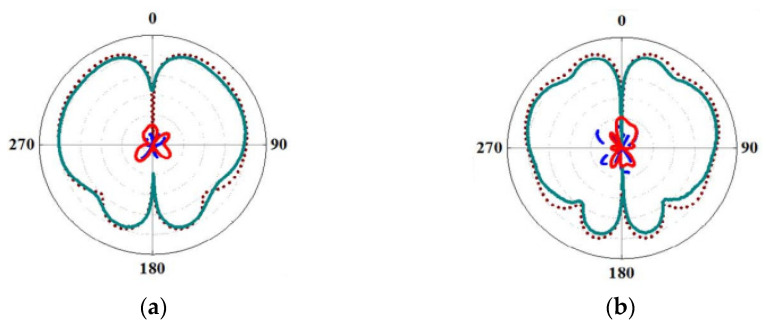
Conical radiation pattern of the proposed antenna in E-plane at: (**a**) 33 GHz; (**b**) 41 GHz [116].

**Figure 29 micromachines-13-02178-f029:**
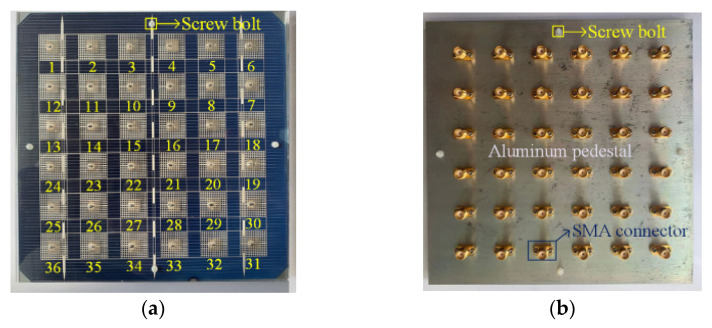
Proposed antenna integrated with solar cell: (**a**) top view; (**b**) bottom view [122].

**Figure 30 micromachines-13-02178-f030:**
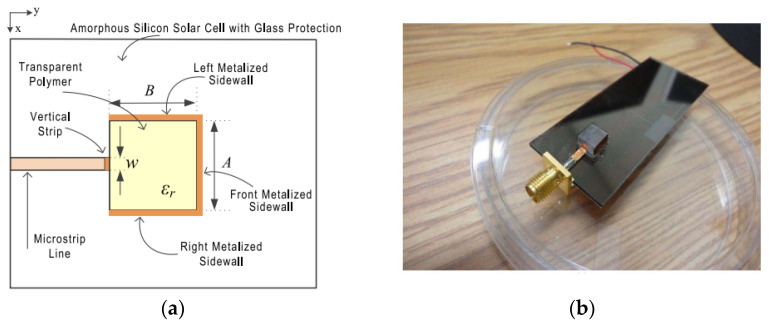
Transparent resonator antenna on silicon solar cell: (**a**) top view; (**b**) fabricated prototype [123].

**Table 1 micromachines-13-02178-t001:** Advantages and disadvantages of DRAs.

Advantages	Disadvantages
DRAs offer a high degree of flexibility over a wide range of frequency bands, thus making it practical for many applications and requirements. DRAs have a small dissipation loss as they feature a high dielectric constant with no conducting parts. Therefore, DRAs can handle high power. DRAs consist of ceramic material. Ceramics offer an excellent stability of temperatures. Therefore, they enable the DRA to operate in a wide temperature range. DRAs have simple geometries such as rectangular, cylindrical, and hemispherical, which are readily available on the market and can be easily fabricated. Various modes can be excited within the DRA element depending on the shape of the resonator. Various modes will generate different radiation patterns for different coverage requirements. The excitation method of DRA is simple and easy to integrate with current technologies. DRAs have a much wider impedance bandwidth compared to microstrip antennas because the DRA radiates over the entire antenna surface except for the grounded part. The operating bandwidth and radiation characteristics of DRAs can be varied by choosing a suitable dielectric constant and the dimensions of the resonator. DRAs can be used at microwave frequencies and higher such as terahertz (THz).	DRAs are problematic for design with a specific frequency compared to microstrip antennas. It is difficult to form and modify a complicated-shaped DR in order to compensate for manufacturing tolerances or fabrication errors. Due to the proximity of resonant frequencies of various modes, the resonant frequencies and the field patterns are not only for the desired modes of operation but also for other undesired modes. DRAs require precise alignment and assembly of the DR because any marginal misalignment between the DR and the feeding network will affect the antenna’s performance due to the shorter wavelengths and the smaller antenna size.

**Table 2 micromachines-13-02178-t002:** Comparison between different single-fed circularly polarized DRAs.

Ref.	Basic Geometry	*f_r_* (GHz)	CP BW (GHz)	CP BW (%)	*ε_r_*	Feed Type	Techniques	Gain (dBi)	Mode	Eff. (%)
[42]	RDRA	3.67–4.73	3.67–4.4	19.9	10	Metal feeding strip	H-shaped conformal metal strip	6.8	${\mathrm{TE}}_{\delta 13}^{x}/{\mathrm{TE}}_{1\delta 3}^{y}$	NM
[43]	CDRA	2.97–4.52	3.07–4.15	29.91	9.8	Microstrip line	Modified ground plane	2.84	HEM_11δ_	94.69
[44]	L-shaped	2.9–3.8	2.97–3.35	42	30	Microstrip line	DGS and PIN Diode	3.4	NM	93
[45]	Truncated RDRA	2.08–2.98	2.31–2.72	17.75	6.6	Coaxial feed	Fluidic dielectric	5.5	TE_δ11_	70
[46]	CDRA	4.3	4.2–4.55	8.3	9.8	Coaxial feed	Stacked DRA	5	NM	NM
[47]	CSDRA	4.45–7.3	5.47–6.37	16.45	9.8	Coaxial feed	Sector DRA	5.8	TM_v1δ/_TM_2v1δ_	80–94
[48]	HDRA	4.16	4.2–5.3	20.9	4.3	Coaxial feed	Fractal geometry	6.38	NM	93
[49]	RDRA	10–13	10.4–11.44	10	10	Aperture Coupling	Two-layer RDRA	11.1	TE_11,11_	NM
[50]	CDRA	20.1–28.2	20.3–26.45	30.3	10.2	Aperture Coupling	Cross-slot, substrate integrated	8.15	HE_11δ/_HEM_12δ+1_	NM
[51]	SDRA	2.22–3.5	2.22–3.72	67.57	10	Aperture Coupling	Metal plate	4.73	TE_111_	97
[52]	Stacked RDRA	19.7–21 27.5–31.2	19.8–20.8 28.7–29.9	5.2 4.1	2.2 10.2	Aperture coupling	Shared-aperture DRA, slot-dipole	6.6 8.2	TE_111_	NM

**Table 3 micromachines-13-02178-t003:** Comparison between different dual- or multi-feed circularly polarized DRAs.

Ref.	Basic Geometry	*f_r_* (GHz)	CP BW (GHz)	CP BW (%)	*ε_r_*	Feed Type	Techniques	Gain (dBi)	Mode	Eff. (%)
[28]	Bowtie-DRA	1.95–2.56 2.93–3.33	1.95–2.56 2.93–3.33	31.3 13.65	10	Microstrip line	Sequential rotated	3.82	${\mathrm{TE}}_{111}^{x}$/${\mathrm{TE}}_{111}^{y}$	83.9
[56]	RDRA	5.769–5.913 5.788–5.922	5.766–5.911 5.788–5.922	2.48 2.28	10	Dual-port	Slot-coupling, SR feeding	8.4 8.2	${\mathrm{TE}}_{\delta 21}^{x}$/${\mathrm{TE}}_{1\delta 1}^{y}$	82 81.5
[57]	CDRA	2.23–3.22	2.18–3.07	40.83	9.9	Probe	Tunable feed network	6.44	HEM_11δ_	NM
[58]	RDRA	1.7–3	1.83–2.85	55.74	9.2	Series	Orthogonal modes	8.4	NM	85

**Table 4 micromachines-13-02178-t004:** Comparison between different feeding configurations.

Feeding Type	Fabrication	Reliability	Impedance Matching	Advantages	Disadvantages
Metal feeding strip	Easy	Better	Easy	Low cost High gain Improvement in bandwidth without any multi layering or complicated design of the DRA	Fabrication process need to be performed carefully to avoid a difference between the simulation and measurement results
Microstrip-line feed	Easy	Better	Easy	Low cost Easy mounting on element	Low Q factor As the thickness of the substrate increases, surface wave and spurious radiation increases which limit the bandwidth Direct coupling between the DR and the line feed will affect the radiation performances of DRA The radiation from the feed line will increase the cross-polar level For mm-wave frequencies, the size of the feed line is similar to the patch size, which leads to an increase in the undesired radiation
Coaxial feed/probe feed	Soldering and drilling	Poor due to soldering	Easy	Can be placed at any desired location inside the patch Easy to fabricate Low spurious radiation	Narrow bandwidth Difficult to fabricate because a hole has to be drilled into the substrate Matching problem for thicker substrate because the probe length will increase and the input impedance will become more inductive
Aperture coupling	Difficult and alignment required	Good	Easy	No physical contact between the feed and radiator Wide bandwidth Allows independent optimization of antennas and feed network by using thickness and permittivity of substrate	Requires multilayer fabrication and alignment is crucial for input matching
SP feeding network	Difficult	Good	Medium	Low cost Improved the performances of array antenna	Difficult to design a sequence feed network
Printed gap waveguide	Difficult	Good	Medium	Compact size Low loss Low dispersion device Does not require a conductive connection between the upper and lower plates	Narrow bandwidth

**Table 5 micromachines-13-02178-t005:** Comparison between various MIMO circularly polarized DRAs.

Ref.	Geometry	*f_r_* (GHz)	CP BW (GHz)	CP BW (%)	*ε_r_*	Feed Type	Techniques	Gain (dBi)	Mode	Isolation (dB)
[65]	RDRA	3.45	3.15–3.93	24.76	9.8	Microstrip line	EBG	4.83	TE_111_	−26
[66]	CDRA	3.22–3.72	3.34–3.54	5.99	9.8	Microstrip line	Polarization and pattern diversity	>4.2	HEM_11δ_	>15
[67]	SDRA	2.4	2.39–2.54	4.9	9.5	Microstrip line	Decoupling. Metal strips on the DRAs	5.2–5.6	TE_111_	31
[68]	CDRA	2.9–3.2 3.44–3.64 4.75–5.5	3.32–3.58 5–5.32	7.8 6.4	NM	Aperture coupled feeding	Orthogonal orientation of antenna elements	>2	${\mathrm{HEM}}_{11\delta }^{x}$/${\mathrm{HEM}}_{11\delta }^{y}$	>15
[69]	V-shaped DR	4.89–5.42	5.16–5.38	4.17	9.8	Microstrip line	Orthogonal feeding network	5.04	TE modes	≤−14
[70]	CDRA	3.38–3.8	3.4–3.57	5	9.8	Microstrip line	Decoupling network	4.91	HEM_11δ_	21
[71]	RDRA	3.5–4.95	3.58–4.4	23	10	Conformal metal strip	Parasitic patch and diagonally position	6.5	${\mathrm{TE}}_{\delta 13}^{x}$/${\mathrm{TE}}_{1\delta 3}^{y}$	>28
[72]	Pyramid DRA	7.74–12.39	8.86–9.69	9.37	10.2	Microstrip line	Symmetry antenna elements	7	Multiple modes	>21.8

**Table 6 micromachines-13-02178-t006:** Comparison between various mm-wave circularly polarized DRAs.

Ref.	Geometry	*f_r_* (GHz)	CP BW (GHz)	CP BW (%)	*ε_r_*	Feed Type	Techniques	Gain (dBi)	Mode
[76]	Tooth-shaped DRA	29.40–34.66	29.75–34.15	13.77	10.2	PGW	SR method	8.33	${\mathrm{TE}}_{113}^{x}$/${\mathrm{TE}}_{113}^{y}$/${\mathrm{TE}}_{115}^{y}$
[77]	CDRA	57–64	55–64.5	17.27	NM	Aperture-coupled feeding	SR method	11.43	HEM_11δ_
[78]	Flower-shaped DR	27–38	29.2–30.7	5.14	10.2	Sequential feeding network	Cross-slot coupling	9.5	NM
[79]	RDRA	28.6–27.26 24.04–26.88	23.54–24.04	2.12	10	Microstrip line	Aperture-coupled slot	6.53	TE_1y1_
[80]	CDRA	25.1–30.1	27.2–29.2	7.35	7 and 6	Coaxial feed	DRA loaded with parasitic dielectric cylinder	6.5	${\mathrm{TE}}_{113}^{x}$

**Table 7 micromachines-13-02178-t007:** Comparison between various DRAs for RF energy harvesting.

Ref.	Geometry	*f_r_* (GHz)	BW (%)	*ε_r_*	Antenna Dimension (*W* × *L* × *h*)λ_c_^3^	Gain (dBi)	Feed Type	Polarization
[85]	Slot-fed RDRA	5–6.04	18.8	7.75	(0.72 × 0.72 × 1.23)	6.69	Slot feed	LP
[86]	Hybrid RDRA	1.67–6.7	120	10.2	(1.25 × 1.11 × 0.13)	9.9	Inverted T-shaped feed line	LP
[58]	RDRA	1.83–2.85	43.59	9.2	(0.55 × 0.55 × 0.32)	7.7–8.4	Series feeding	CP
[87]	Quarter sectored CDRA	2.28–2.75	18.7	NM	(0.33 × 0.31 × 0.06)	4.3	Microstrip feed	CP
[88]	Quarter sectored CDRA	5.75–5.85	1.7	10	(0.68 × 0.68 × 0.22)	7.02	Aperture slot feed	CP

**Table 8 micromachines-13-02178-t008:** Comparison between various DRAs for RFID.

Ref.	Geometry	*f_r_* (GHz)	BW (%)	*ε_r_*	Gain (dBi)	Feed Type	Polarization
[89]	Cylindrical DR	0.86–0.92	6.74	100	1.61	Discrete port	LP
[90]	Cylindrical DR	0.85–0.9	5.71	506	7.5	Discrete port	LP
[91]	Cylindrical DR	3.5–4.5	25	35	NM	Coaxial port	CP
[92]	Crossed split-ring resonator	0.907–0.914	0.77	NM	5.51	Coaxial port	CP
[93]	Disk-type cylindrical DR	4–7	54.55	35	14	NM	LP
[94]	Cylindrical DR	Between 100–200	NM	35	16.7	NM	LP

**Table 9 micromachines-13-02178-t009:** Comparison between various DRAs for radar applications.

Ref.	Geometry	*f_r_* (GHz)	BW (%)	*ε_r_*	Gain (dBi)	Polarization	Applications
[95]	Rectangular DR, array	3.79–6.29	49.6	10.2	18–22	LP	Unmanned aerial system
[96]	Rectangular DR, array	23.25–24.93	6.97	16	9.5	LP	Short-range vehicular radar
[97]	Cylindrical DR	22.5–26	14.43	9.9	10.8–11.8	CP	Automotive radar
[98]	Cylindrical DR	8.2–10.3 (Element) 8.2–11.3 (Array)	22.7 31.79	20 20	7.7 9.5	CP	Weather monitoring and vehicle speed detection

**Table 10 micromachines-13-02178-t010:** Comparison between various DRAs for biomedical applications.

Ref.	Geometry	*f_r_* (GHz)	BW (%)	*ε_r_*	Gain (dBi)	Polarization	Applications
[100]	Hemispherical DR	2.4–2.5	4.08	22	−15	DP	WCE
[101]	Rectangular DR	2,4–2.5	4.08	80	−23.6	LP	Implantable medical devices
[102]	Rectangular DR	7.47–8.25	9.92	10	5	CP	WBAN
[103]	Cylindrical DR	2.46–2.59	5.15	12.85	NM	CP	Breast cancer detection

**Table 11 micromachines-13-02178-t011:** Comparison between various DRAs for vehicular applications.

Ref.	Geometry	*f_r_* (GHz)	BW (%)	*ε_r_*	Gain (dBi)	Polarization	Applications
[104]	Rectangular DR	1.7–2.2 2.5–2.7	25.6 7.69	9.2	4	LP	LTE automotive
[105]	Cylindrical DR	3.3–3.6 10–10.5	8.7 4.88	10	5.8 12.0	LP	Short-distance vehicle-to-base station and long-distance vehicle-to-satellite communications
[116]	Cylindrical DR	3.1–5.6	57.5	9.8 16	8.7	LP	Vehicle-mounted surveillance equipment
[52]	Stacked Rectangular DR	19.8–20.8 28.7–29.9	4.92	2.2 10.2	6.6-8.2	CP	UAV
[66]	Cylindrical DR	3.34–3.54	5.8	9.8	42	CP	V2V to V2I communications
[117]	Cross-shaped DR	6.26–8.74	33.1	6.15	10.77	CP	Aerial vehicle

**Table 12 micromachines-13-02178-t012:** Comparison between various transparent antennas for solar cells.

Ref.	Technology Used	*f_r_* (GHz)	BW (%)	Gain (dBi)	Eff. (%)
[120]	ITO copper	25–27	7.69	22.2	NM
[121]	ITO copper	8–12	40	17	65
[122]	Metal mesh	8.51–9.1	6.7	20.14	>35
[123]	Transparent DR	9.1–9.7	6.4	6	NM

## Data Availability

Not applicable.

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
