# Peer review of "A Review of Circularly Polarized Dielectric Resonator Antennas: Recent Developments and Applications"

_micromachines, 2022, doi:10.3390/mi13122178_

Round 1
Reviewer 1 Report
In this paper, authors presents a review on recent developments and applications of dielectric resonator antennas (DRAs) is proposed in this paper.
The information given in this review paper is useful for researchers who are working on DRAs.
This is a preview article, as for this type of article it is written correctly and can be published after introducing a few editorial changes.
Several elements need to be corrected in the paper:
(1) Please improve the quality of the figure 1, figure 4, figure 9, figure 11, figure 16.
(2) Authors can add information on the difference between linear and circular polarization (preferably in the form of a figure).
(3) The authors can add an example of an antenna used in solar cells (preferably in the form of a drawing as in other cases).
Reviewer 2 Report
· · The paper is mainly focused on circularly polarised DRA and this needs to be mentioned in the title.
· In the abstract, the comment “and Multiple-Input-Multiple-Output (MIMO) used by different researchers for generating circular polarization” is incorrect as MIMO utilises circularly polarised DRA elements to start with and has nothing to do with generating the circular polarisation.
· In the abstract, the comment “innovative design solutions for broadening the circular polarization bandwidth and reducing the mutual coupling are investigated”, are the authors presenting a review article or investigating novel DRA designs?
· Line 37, “and suitable for millimeter-wave applications”, does this comment refers to the DRA in [2] or DRAs in general? If it is the latter, then it is out of place and need to be discussed in more details.
· Line 56, “Linearly polarized (LP) DRAs have been explored broadly up until the early 2000s”. Actually, LP DRAs are still receiving considerable interest as can be verified from recent publications in the literature.
· Line 66, the comment “over a wide bandwidth and wide beamwidth”; not necessarily correct as numerous DRAs have been reported for DRAs with narrow bandwidth and beamwidth.
· Line 76-77, a reference needs to be cited to support this point. I cannot see how a CP can be achieved simply by “extending the far end of the microstrip vertically to the cylindrical surface of the DRA”
· Line 78-79, an AR bandwidth of less than 10% cannot be classified as a narrow CP bandwidth.
· Line 82, why you need a polarizer if the CP is achieved using dual feeding?
· DRAs and DRA acronyms are the same, there is no need to mention them individually.
· The acronym (DR) has been defined twice in lines 69 and 111.
· Line 97, “comprehensively explains about several applications using DRA” this comment needs rephrasing.
· Line 110; “It works by feeding the transmit signal into the resonator material [23].” Do you really need this comment as it is a common knowledge. What is a resonant material?
· Line 113-114, if we avoid the degenerate modes, then how the CP radiation can be achieved?
· Lines 130-131, this comment has been mentioned already.
· Table I, too many DRAs have been reported at operating frequencies that are significantly higher than 44 GHz, e.g THz. In addition, one of the key disadvantages of the DRA is the required assembly and alignment, which has not been mentioned.
· Line 147; “a narrow AR bandwidth up to 20%”, the authors need to think about their definition of narrow AR bandwidth, which is usually defined as ~3% or less.
· Line 155, “offer a planar configuration”, with the DRA there is no planar configuration to speak off as the whole structure is 3D by nature.
· Microstrip line feed has limitations such as the direct coupling between the DRA and the feed network as well as the possible radiation from the feed network. As mentioned earlier, the authors need to cite references that support their arguments with respect to the feed network choices.
· Table 2, please check the feeding methods as there are papers using aperture coupling, which has been mentioned as a microstrip line feed.
· Line 324, did they use a “patch antenna” in [65]?!
· Line 374, what is meant by “a dual CP DRA”?
· Table 6, is this for mmwave DRAs? The comparison represents personal opinions of the authors by using subjective terms such as “easy” “better”, etc ..
· Lines 575-576, such a claim needs to be justified by citing relevant references as mentioned earlier. In addition, microstrip line feed has several limitations and it maybe totally unsuitable at the mmwave frequencies.
Reviewer 3 Report
Please see the attachment.

Reviewer 4 Report
1) Review on DRA modes, Antenna parameter enhancement technique like bandwidth, gain etc. has not been done in the paper.
2) Different types of analysis method is not explained in the paper like Dielectric waveguide model, Bessel function, Green method etc.
3) Formula for calculation of resonant frequency, quality factor of each modes has not been done in the paper.
4) For the excitation of antenna different types of feeding method is used is not explained in the paper and also comparison among them is not done in the paper.
5) Low-profile, Compact DRA, DRA Arrays and Conformal antenna is also not explained in the paper.
6) Applications of the DRA is properly explained in the paper.
7) There are some other works in DRAs which authors have not reported including fractal geometry investigations. Can the authors add a small subsection in wide band area and include some comments about it.
Round 2
Reviewer 2 Report
The authors have addressed all my comments.
Reviewer 3 Report
It is OK by me now.
Reviewer 4 Report
The paper can be accepted.